# WetBench: LLM-Based Simulation Environment to Evaluate Wet-Lab Experiment Planning and Design

## Abstract

We introduce WetBench, an LLM-based simulation environment for scalably evaluating AI systems' ability to design and plan wet-lab experiments. Traditional evaluation approaches are limited by the expense and safety concerns of executing AI-generated experiments in physical laboratories. To address this, we developed a simulation environment that uses LLMs as state transition models to simulate experimental outcomes and as evidence classifiers to evaluate whether experiments provide sufficient information to achieve stated goals. WetBench includes 18 expert-curated experimental configurations spanning cell biology, neuroscience, microbiology, and analytical chemistry, each validated as solvable within the environment's constraints. We evaluated the fidelity of our LLM-based simulation through expert ratings, finding that state transitions were judged as highly plausible (90% plausibility) by human expert raters. Evidence classification showed substantial agreement between LLM classifiers and human experts (72-82% agreement), on par with inter-human baseline agreement (75%). Using this environment, we benchmarked frontier language models on experimental design and planning capabilities. GPT-5 demonstrated superior performance with a 44.4% pass@1 rate that increased to 72.2% at pass@5, substantially outperforming other models, including Gemini 2.5 Flash (50.0% pass@5), Qwen 3 (41.2% pass@5), and Claude Sonnet 4 (27.8% pass@5). We open-source WetBench as a Python gymnasium environment to support further development of AI systems for autonomous scientific experimentation. [1]

## 1 Introduction

Artificial intelligence has significant potential to accelerate research in biology and bioengineering. While recent work has demonstrated substantial progress in using AI systems for hypothesis generation ( Gottweis et al.; Baek et al.) and data analysis ( Aygün et al.), a critical component of the discovery process in biology remains underexplored: experimental planning and design. Historically, large language models (LLMs) have been criticized for weak planning capabilities: they often generate plans that are not executable, lack the ability to self-verify or refine them, and struggle to manage complex constraints ( Kambhampati et al.; Xie et al.; Vyas et al.; Chang et al.). Additionally, while there is work suggesting reasonable human-LLM agreement in hypothesis evaluation ( Ghareeb et al.), this has not been established in the context of evaluating experimental design. Progress in this area is critical, as we may soon be bottlenecked not by the number of scientific hypotheses AI systems can generate, but by our ability to design and execute experiments to evaluate them ( Reddy & Shojaee; Zhang et al.).

While successful experimental planning and design requires many features, we focus on two in particular:

- **Feasibility** A feasible experiment consists of a set of steps that can be performed in the lab, given constraints on materials, equipment, experimental protocols, and realistic expectations.

---

[1]We will provide the link to the GitHub after de-anonymization

- **Informativeness** An informative experiment provides the information required to answer the intended question through well-designed controls and conditions.

Evaluating the ability of AI systems to develop feasible and informative experiments in the physical world is challenging for multiple reasons. Having human experts execute every LLM-generated experiment in the lab is prohibitively expensive, slow, and could confound AI experimental ability with human expertise. While many groups have used liquid handlers as a platform to explore autonomous experimental design, liquid handlers significantly constrain the space of potential experiments. On the other hand, providing AI agents with direct, unsupervised access to large, diverse experimental facilities—such as self-driving labs ( Qiu et al.) or cloud laboratories ( Boiko et al.)—raises safety concerns and incurs significant resource and time costs ( Sandbrink). Each approach falls short of providing the scale and diversity necessary to meaningfully evaluate and improve experimental planning capabilities in AI systems.

Considering this, we sought to determine whether LLM-based experimental simulation environments have the potential to serve as surrogates for experiments. Through strong performance on challenging biological reasoning benchmarks like GPQA ( Rein et al.) and Lab-Bench ( Laurent et al.), there is evidence that frontier reasoning LLMs have internalized large amounts of biological and experimental knowledge. If we can use LLMs to realistically simulate experimental procedures and judge when experiments are sufficiently informative, we could provide a sandbox for evaluating LLM experimental design ability and an environment to train AI agents in a way that is scalable, highly diverse, and safe (such as in Liu et al.). Critically, for the system to be useful, we do not need or expect LLMs to predict completely novel experimental outcomes. Experimental design in practice involves executing well-established protocols to investigate questions where outcomes remain uncertain. For example, to test a new biosensor, a researcher generally plans by modifying experimental protocols they are confident in to design the experiment with appropriate controls. While LLMs should not be able to determine if the biosensor would work, they should be able to design feasible experiments using reasonable protocols and informative controls to gather the information required to answer the question.

To explore this idea, we developed WetBench, an LLM-based wet-lab simulation environment. AI agents begin with an experimental goal, initial materials, and available actions (e.g., Combine, Incubate, UseMicroscope). At each step, agents take actions on materials and submit them to a state transition model, which simulates resulting material property changes and physical observations. After accumulating a history of actions and observations, agents submit their evidence to a classifier that determines whether the cumulative results sufficiently justify the experimental goal. We manually curated a set of experimental configurations (i.e., an experimental goal and initial materials) to serve as a benchmark for the WetBench environment. After verifying that each experiment was solvable within the constraints of the environment, we had AI agents attempt to solve these problems and used the resulting state transitions and evidence submissions to evaluate the system's fidelity. Our work extensively examines when these LLM-based state transitions and evidence interpretations align with expert judgment and identifies systematic failure modes.

Our work makes five key contributions: (1) we develop a simulation environment for wet-lab experiment execution; (2) generate an expert-verified experiment benchmark for evaluation; (3) evaluate the fidelity of LLM-based state transitions and evidence classification using expert ratings; (4) explore the current abilities of frontier and open-source models on experimental design within the environment; and (5) open source the project to support further development of both the environment and future AI agents.

## 2 RELATED WORK

A complementary line of work focuses on translating human-composed protocols into structured instructions. BioPlanner ( O'Donoghue et al.) converts natural language protocols into pseudocode for laboratory automation, utilizing a teacher-student framework. In their approach, a "teacher" LLM model converts existing protocols from the literature into pseudocode for execution in a robotic lab. A "student" model is then evaluated on its ability to reconstruct this pseudocode when given only the protocol description and admissible pseudofunctions. The student's pseudocode is then evaluated based on next-step prediction, full protocol generation, and pseudofunction retrieval, with success measured against the teacher model's pseudocode. While this approach enables automatic

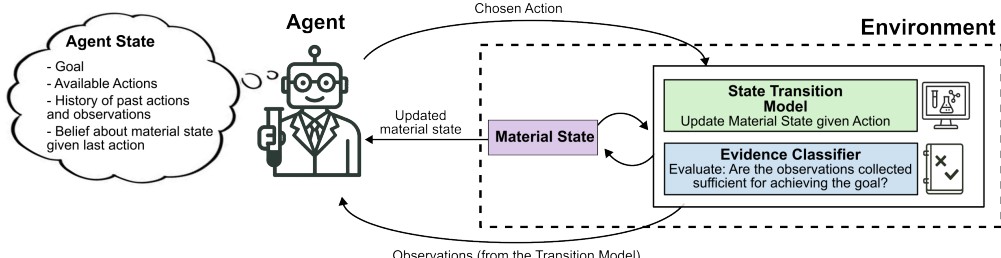

Figure 1: **Overview of the WetBench system**. The AI agent maintains goals, prior history, and knowledge of the available materials and actions. At each step, the agent chooses an action to make progress towards its goal. If the action is a *transformation*, *measurement*, or *simulation* action, it is submitted to the state transition model that determines the result of new materials and observations. If the agent chooses "Submit" then all prior actions and observations are sent to the evidence classifier to determine if the goal was achieved.

evaluation of protocol conversion accuracy, it addresses the translation of known experimental procedures rather than the design of novel experiments to achieve informational goals. Additionally, their framework does not model experimental outcomes or assess whether proposed actions would actually succeed in achieving the scientific objectives. In contrast, our work addresses open-ended experimental planning, where models must determine which experiments to perform based on scientific objectives, with success defined by the sufficiency of evidence for target interpretations rather than adherence to predetermined procedural sequences.

Coscientist ( Boiko et al.) demonstrated autonomous experimental planning, using LLMs with tool access, including web search, code execution, documentation retrieval, and direct control of robotic lab equipment through APIs. The execution through robotic lab systems (such as Opentrons and Emerald Cloud Labs) defines a fixed action space that constrains the plans of the LLM coscientist. However, in this system, the goals given to the coscientist are always assumed to be feasible given the resources and action space, ignoring the important roles of feasibility and evidence sufficiency evaluation in planning.

## 3    WETBENCH SYSTEM OVERVIEW

The WetBench system consists of an environment and an agent. The environment is comprised of the material state, associated observations, a set of experimental actions, a state transition model that simulates these actions, and an evidence classifier that determines whether the series of materials, actions, and observations constitutes sufficient evidence for the experimental goal.

The agent operates with an experimental goal and knowledge of available actions and materials in the environment. The agent's task is to design and execute an experiment in the environment that is sufficiently informative to achieve the experimental goal. These components interact in a closed-loop fashion: the agent attempts to execute actions in the environment; the state transition model either rejects the action (due to it being infeasible or outside of the set of available actions) or simulates the effect of the action by modifying the relevant materials, producing physical observations, and stepping in time. After receiving the observation and materials changes, the agent chooses experimental actions until it believes it has reached its goal. When the agent believes its actions and observations are sufficient for the goal, it submits the results to the evidence classifier, which accepts or rejects them based on whether they are sufficiently informative.

### 3.1    AGENT

The agent is an LLM-based system prompted to achieve specific experimental goals using a defined set of available materials and actions. The agent receives an experimental objective, an initial inventory of materials, and knowledge of the available action space. For instance, the goal might be to determine calcium response dynamics in cortical neurons following KCl depolarization, starting with materials like specific plasmids, stock solutions, and neuronal cultures.

The agent's decision-making process relies entirely on the LLM's reasoning capabilities and scientific knowledge, without access to external tools or databases beyond what is explicitly provided in the simulation environment. This design ensures that experimental planning performance reflects the model's intrinsic understanding of laboratory procedures and scientific methodology.

## 3.2 Material state and actions

**Material state**  The material state defines what the agent has access to conduct its experiments. Typical materials include biological samples (e.g., cell lines, primary neuron cultures, etc.), reagents (e.g., buffers, growth media, etc), containers (e.g., Eppendorf tubes, 50 mL conical centrifuge tubes, etc.), and chemicals (e.g., hydrochloric acid, sodium hydroxide, ethanol, etc.). Each material is represented using a structured dictionary format with a name, physical/chemical/properties properties (e.g., concentration, container specifications, cell type, etc.), and a unique barcode identifier (Figure 2, Top Left). Various addition properties are included to enforce realism in the environment. Each material has a defined environmental condition in which it exists (e.g., held at room temperature, in a 4°C refrigerator, or -80°C freezer, etc). Finally, each material is labeled as static or dynamic. Dynamic materials are materials that dynamically change over time; for example, cells that continue to grow or chemical reactions that require a certain amount of time to complete. The age of the material is tracked through the "created_at" and "last_modified" properties. As time passes in the simulation environment, the material state changes based on the environmental conditions it is in and its properties.

**Actions**  At each step, the agent chooses an action to apply to a subset of materials to make progress towards the goal. The set of actions was designed to cover the set of typical actions done during wet-lab experiments (detailed in Appendix C). Some actions like "Transfer," "Incubate," or "Wash" are *transformation* actions as they primarily change the physical properties of the materials. Other actions like "UseMicroscope"or "MeasurepH" are *measurement* actions because they primarily produce data and observations (they may or may not affect the physical properties of the material). Finally, there are *simulation* actions like "Wait" that allow the agent to interact with its simulation environment directly (e.g., stepping forward in time). Each action has required and optional parameters with fixed constraints on the parameter values (Figure 2, Bottom Left). This is to prevent the agent from inventing action parameters to "hack" the system into producing any output it wants. [2]

## 3.3 State Transition Model

When the agent chooses an action, parameters, and target materials, this information is submitted to the State Transition Model (detailed in Appendix C.2). The State Transition Model is an LLM prompted to decide if the action should be rejected and, if not, to simulate plausible outcomes of the action. The rejection reasons are not limited to, but include:

- Missing critical materials or reagents
- Insufficient amounts or values (e.g., the new container intended to hold the sample is too small, the measurement device cannot reliably record a signal in liquid below a certain volume)
- Incorrect material state (e.g., frozen samples needing to be in a liquid state, adherent cells needing to be in suspension)
- The described action is not valid considering the set of available actions and parameter constraints.

When simulating plausible outcomes, the state transition model is prompted with a detailed description of how the action would be conducted in the physical world and instructed to consider:

- Chemical reactions and physical processes
- Conservation laws and material balance (i.e., what goes in must come out)

---

[2]"VisualInspection" is the only action that has a free-string parameter (it is constrained to be a regular expression that defines a single sentence that ends with a question mark). However, the state transition model is prompted to reject any inquiries that cannot be ascertained by looking at the sample.

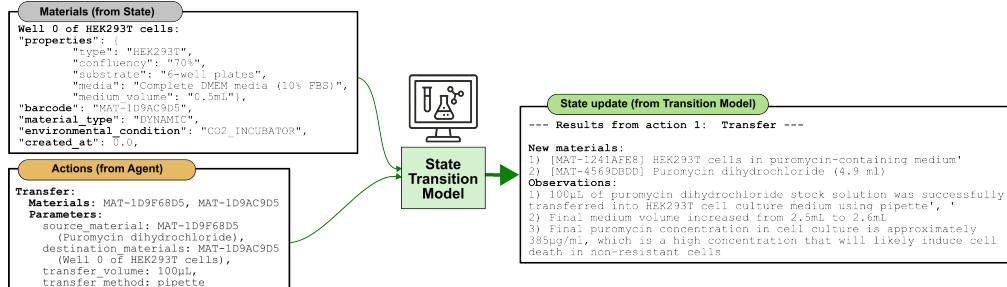

Figure 2: **State transition model workflow.** Upper left: Example of individual material in material state with a set of physical properties, a unique barcode, and environmental information. Another material (puromycin) is omitted due to space constraints. Lower left: Example of action. The target materials refer to the material that goes into the action (e.g., HEK293T Cells and puromycin). The action has associated parameters including the source and destination materials, the transfer volume, and the transfer method. Right side: The outcome of a state transition model for the action. Two new materials are generated: HEK293T cells with puromycin and residual puromycin. Additionally, the state transition model generates a set of physical observations corresponding to the action and the new materials that have been introduced.

- Realistic experimental outcomes
- Time-related processes (growth, decay, transformation, etc.)
- Observational limits (what could be observed by a human or AI with human-like sensory capabilities.)

The state transition model outputs a structure dictionary that describes the resulting material state and a set of associated observations from the experimental action. It is through these constraints on the state transition model that we work to force the agent into producing **feasible** experiments.

### 3.4 EVIDENCE CLASSIFIER

Within the environment, an experiment can end for one of three reasons:

- The agent believes it has sufficient evidence for its goal and executes the "Submit" action, which submits the history of observation to the evidence classifier.
- The agent does not believe it can accomplish its goal and executes the "Quit" action
- The agent attempts the maximum number of allowed actions (default 100 actions), or three actions are rejected in a row

The evidence classifier evaluates whether cumulative experimental evidence supports the stated goal by receiving a comprehensive experimental history, including all actions, materials, timing, and collected observations, organized chronologically. This evidence history is evaluated against the experimental goal. The classification prompt instructs the model to consider scientific plausibility, logical consistency, appropriate inference levels, alternative explanations, evidence strength, and confounding factors in its reasoning (detailed in Appendix C.3).

For example, in an experiment aimed at determining calcium response dynamics in cortical neurons following KCl depolarization, sufficient evidence might include: baseline calcium measurements, controlled KCl application with appropriate concentrations, time-series calcium imaging data showing clear signal changes, and negative controls without KCl treatment. Insufficient evidence might consist of only a single calcium measurement without controls, unclear timing of KCl application, or missing baseline measurements that prevent interpretation of any observed changes. Critically, it is through the evidence classifier that we enforce the **informativeness** requirement of successful experimental design. Just as the state transition model constrains agents to produce feasible protocols by rejecting impossible actions, the evidence classifier ensures that agents must gather sufficient evidence—with appropriate controls and logical inference—to meaningfully address their experimental goals, rather than simply executing a series of plausible-sounding steps.

## 4 METHODS

### 4.1 EXPERIMENT CONFIGURATIONS

For the benchmark, we generated 18 experimental configurations spanning cell and molecular biology, neuroscience, microbiology, and analytical chemistry. Each experimental configuration is defined by a set of materials and a scientific goal. We curated the experimental configurations based on experiments that the authors have run in the wet-lab or based on the literature. The full list of experiments is in the (detailed in Appendix B).

### 4.2 MODEL CONFIGURATION

We evaluated multiple models as experimental design agents:

- **Proprietary reasoning models:** Claude 4 Sonnet (extended thinking), GPT-5, Gemini 2.5 Flash
- **Open-source reasoning models:** Qwen 3 (235B), DeepSeek R1
- **Non-reasoning model:** GPT-4o

For the WetBench environment components, we used Claude 4 Sonnet as the state transition model and GPT-5 as the evidence classifier. Complete model parameters are provided in the Appendix A.

### 4.3 EXPERT RATINGS

For each evaluation task, we selected multiple Ph.D. or postdoc-level researchers in biology, bioengineering, or neuroscience. We instructed each evaluator on the specifics of the tasks and reviewed two to three examples prior to the ratings. Each rating generates a binary score (either plausible or not plausible for state transitions and sufficient or not sufficient for evidence classification), as well as a confidence measure from one (very low) to five (very high).

## 5 RESULTS

### 5.1 STATE TRANSITION MODEL VALIDATION

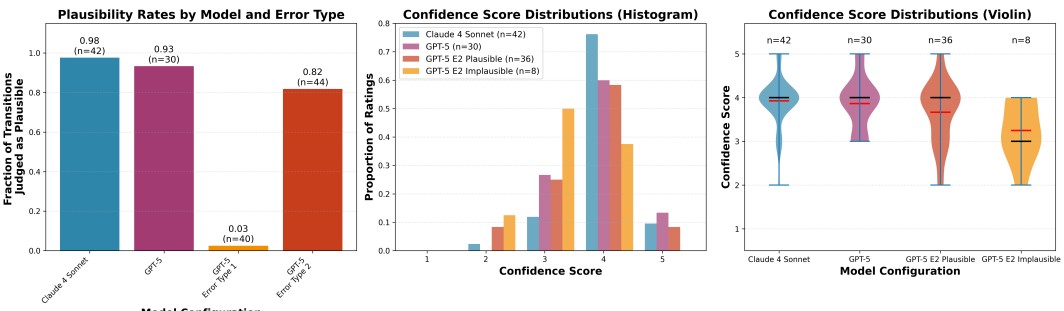

Figure 3: **Expert validations of state transition model.** Left Panel: Across ratings, we find overwhelming agreement that the state transition tests are plausible for Claude 4 Sonnet and GPT-5. Experts rate GPT-5 error type 1 transitions extremely low and rate GPT-5 error type 2 at lower but similar plausibility rates to the unperturbed models. Center Panel: Histogram distribution of the Claude 4 Sonnet confidence ratings, GPT-5 confidence ratings, and GPT-5 error type 2 confidence ratings split into ratings that were deemed plausible and implausible. Right Panel: A violin plot comparing Claude Sonnet, GPT-5, and GPT-5 error type 2 split into plausible and implausible.

To understand the performance of different LLMs as state transition models, we collected a large set of (input materials, action) pairs from multiple simulation runs and used Claude 4 Sonnet and GPT-5 to generate state transitions. We then collected plausibility ratings from evaluators, who were

Table 1: Inter-rater agreement metrics between human-model and human-human

| MODEL | AVG KAPPA | AVG AGREEMENT |
|---|---|---|
| Claude 4 Sonnet | 0.618 | 81.9% |
| GPT-4o | 0.583 | 80.4% |
| Gemini 2.5 Flash | 0.472 | 74.7% |
| GPT-5 | 0.437 | 72.2% |
| Qwen 3 | 0.430 | 72.2% |
| Human | 0.526 | 75.0% |

prompted to assess whether the outcome materials and observations were plausible given the action and input materials.

**LLM-simulated state transitions rated as highly plausible.** Evaluators found the overwhelming majority of predicted transitions to be plausible (90% for Claude 4 Sonnet, 93% for GPT-5). Confidence was generally high (average of $3.90 \pm 0.59$), and anecdotal reports suggested that both models could produce reasonably plausible outcomes. Unsurprisingly, due to the high agreement on plausibility, inter-rater agreement was quite high (Cohen's $\kappa = 0.811$).

**Raters struggled to identify subtle errors in state transitions.** To calibrate our results, we used GPT-5 to introduce intentional errors into state transitions using two error types: obvious errors (Error Type 1) and subtle but legitimate errors (Error Type 2) (see Appendix C.2). This approach allowed us to evaluate rater performance when transitions were intentionally incorrect across different error sensitivities. While raters easily identified Error Type 1 transitions ( 3% rated as sufficient), their judgments of Error Type 2 transition plausibility were similar to baseline transitions. In our post-hoc analysis, we found that Error Type 2 transitions contained very subtle errors, such as calculation errors in reagent concentration, incorrect magnification objectives for imaging, or adding frozen reagents before thawing. These errors were typically single mistakes embedded within heavily detailed and otherwise reasonable outcomes. Given that evaluators consistently reported that parsing state transitions was challenging due to the cognitive load, they appear reliably able to discern general plausibility but struggle with subtle technical errors.

**Subtle errors are correlated with low confidence ratings** Given the cognitive load of parsing dense state transition information, we examined whether type 2 errors correlated with rater confidence. We compared confidence score distributions between regular predictions and type 2 error predictions. There was a modest relationship between confidence and correctness (AUC = 0.609), though not significant enough to serve as a reliable predictor (Figure 3). This relationship held for both error types regardless of whether they were judged as plausible or implausible. However, transitions judged as implausible showed substantially lower mean and median confidence scores compared to all other conditions.

## 5.2 EVALUATING EVIDENCE CLASSIFICATION AGREEMENT

**Evaluators agree with LLM and human evidence classifiers at similar rates.** To validate LLM-based evidence classification, we compared agreement rates between human expert evaluators and different language models. Claude 4 Sonnet achieved the highest agreement with human evaluators at 81.9% ($\kappa = 0.618$), followed by GPT-4o at 80.4% ($\kappa = 0.583$). Both models exceeded the baseline inter-human agreement of 75.0% ($\kappa = 0.526$). The remaining models showed lower but still substantial agreement: Gemini 2.5 Flash at 74.7% ($\kappa = 0.472$), and both GPT-5 and Qwen 3 at 72.2% ($\kappa = 0.437$ and 0.430, respectively). The narrow range of agreement rates across models (72.2% to 81.9%) and the fact that top-performing models exceeded human baseline agreement suggest that LLMs can serve as reliable evidence classifiers in the simulation environment without requiring extensive human expert evaluation for each assessment.

**Inter-model agreement reveals systematic differences in classification stringency.** Overall agreement on evidence sufficiency among the five models was high, with agreement rates rang-

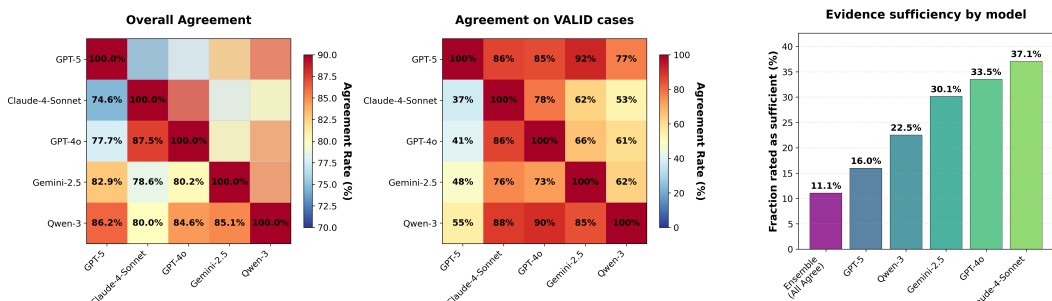

Figure 4: **Inter-model agreement rates.** Left: Agreement matrix between each model for both evidence judged sufficient and insufficient. Middle: Conditional agreement plot showing the rate at which the column model believes evidence is sufficiently valid, conditional on the row model believing that evidence is sufficiently valid. Right: The overall fraction of evidence deemed sufficient across different models and the ensemble. Note: DeepSeek R1 was excluded from this analysis due to persistent errors in its API.

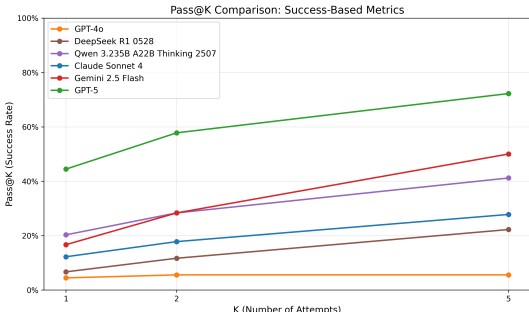

Figure 5: **pass@k rates across all models.** Increasing the number of attempts (K) leads to increased performance across all reasoning models, but does not improve GPT-4o's performance.

ing from 74.6% to 87.5%. However, this is somewhat inflated by overwhelming agreement on cases in which there is insufficient evidence. When examining cases where models disagreed on sufficient experiments, clear patterns emerged in their classification tendencies (Figure 4). GPT-5 demonstrated the most conservative approach, classifying only 16% of evidence submissions as sufficient—the lowest rate among all models tested. In experiments that GPT-5 judged to be sufficient, all other models graded the evidence as sufficient at rates between 77% to 92% (Figure 4, Middle Panel). Conversely, Claude 4 Sonnet was the most permissive classifier, rating more than a third of experiments run as having produced sufficient evidence. When restricting analysis to cases where all models reached consensus on sufficiency, only 11.1% of total classifications were deemed adequate. This suggests significant heterogeneity in what experiments different LLM evidence classifiers believe are sufficiently informative.

### 5.3 FRONTIER MODEL PERFORMANCE IN THE WETBENCH ENVIRONMENT

Using the WetBench environment and benchmark, we evaluated how well frontier language models could accomplish experimental goals across the 18 experimental configurations. Informed by our validation results, we used Claude 4 Sonnet as the state transition model and GPT-5 as the evidence classifier for all agent evaluations. Although GPT-5 did not achieve the highest alignment with human raters, we utilized it to evaluate our metrics due to its overall evidential stringency.

**GPT-5 demonstrates superior experimental design capabilities.** GPT-5 achieved the strongest performance across all metrics, with a pass@1 rate of 44.4% that increased to 72.2% at pass@5 (Figure 5). This substantial performance represents successful experimental design and execution

on nearly three-quarters of the solvable benchmark problems when given five attempts. The model's high baseline success rate of 44.4% indicates consistent experimental planning abilities rather than reliance on multiple attempts.

**Performance stratifies into distinct tiers across model families.** The remaining models showed considerable performance variation, clustering into distinct capability tiers. Gemini 2.5 Flash and Qwen 3 formed a second performance tier, achieving pass@5 rates of 50.0% and 41.2% respectively, though both started from lower pass@1 baselines (16.7% and 20.3%). Claude Sonnet 4 and DeepSeek R1 demonstrated more limited capabilities, with pass@5 rates of 27.8% and 22.2%. Notably, Claude Sonnet 4's constrained performance may reflect its limited thinking budget (2048 tokens) relative to other reasoning models rather than fundamental planning limitations. GPT-4o showed the weakest performance, achieving only a 5.6% pass@5 rate, suggesting that reasoning capabilities may be particularly important for experimental design tasks.

**Qualitative analysis of AI experimental planning and design** To understand how successful AI agents approach experimental design compared to human experts, we examined three representative cases where models achieved their experimental goals (see Appendix B for extended case studies). Our qualitative analysis revealed distinct patterns in AI versus human experimental planning. For protocol-like tasks with minimal control requirements, AI agents converged with human strategies—Gemini 2.5 Flash pursued nearly identical approaches to human experts when determining GFP insert orientation, differing only in minor procedural details. However, when experimental goals allowed multiple valid interpretations, strategies diverged substantially. In characterizing acidic soil isolates, Qwen 3 employed repeated baseline pH measurements against each culture's own baseline, while human experts used uninoculated media controls—both representing reasonable but distinct approaches to the same goal. Most notably, AI agents consistently missed domain-specific shortcuts that experts readily recognized. When determining zebrafish genotypes, GPT-5 defaulted to DNA extraction workflows despite the availability of a simple visual inspection shortcut that human experts employed, highlighting how domain expertise enables efficient experimental design that general reasoning approaches may overlook.

## 6  CONCLUSION

In this work, we explored the potential of using LLMs to construct a simulation environment that promotes high-quality experimental planning and design. Our validation studies demonstrated that LLM-based state transition models produce outcomes that experts rate as highly plausible (90% plausibility ratings), though we identified persistent challenges in evaluating subtle technical errors within information-dense state transitions. For evidence classification, we found that LLM agreement with human experts often matched or exceeded inter-human baselines (81.9% for Claude 4 Sonnet vs. 75.0% human-human agreement), suggesting strong potential for automated evidence evaluation in experimental contexts. Using the WetBench environment to evaluate frontier models revealed substantial performance differences, with GPT-5 achieving the strongest results at 72.2% pass@5, demonstrating successful experimental design on nearly three-quarters of benchmark problems.

The WetBench environment enables rapid and scalable training of experimental design agents through reinforcement learning, offering high-fidelity state transitions and reliable evidence classification. This virtual setting allows agents to iterate through thousands of experimental scenarios, advancing planning strategies that would be impractical to develop in physical laboratories. As hypothesis generation by AI outpaces our experimental capacity, WetBench offers a solution for developing agents that can help close that gap and accelerate biological research.

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

## A    APPENDIX

## B    CASE STUDIES

To understand how successful AI agents approach experimental design compared to human experts, we examined three representative cases where models achieved their experimental goals. These cases reveal both convergent and divergent strategies between AI and human experimental planning.

*Case Study 1:Determine which of 20 colonies contain correctly oriented and in-frame GFP inserts without mutations*

**AI agents converge with human strategies on protocol-like tasks.** Gemini 2.5 Flash pursued a nearly identical approach to the human expert. Both aliquoted multiple media tubes for storage, inoculated GFP colonies, incubated cultures over 24 hours, then extracted and sequenced DNA. The only notable difference was the AI agent's attempt to use microscopy on inoculation tubes, which was rejected due to incorrect tube format (50 mL microcentrifuge tubes). This experiment represents a standard protocol with minimal control requirements—only a contamination control tube—making convergent strategies unsurprising.

*Case Study 2: Characterize which acidic soil isolates achieve the greatest pH reduction when grown on glucose substrates*

**Different interpretations of experimental goal lead to divergent but valid approaches.** Qwen 3 initially followed human strategy by diluting and resuspending soil samples, though using acetate buffer instead of water for inoculation. However, strategies diverged substantially after this point. The human design relied on inoculating multiple distinct microbial colonies with uninoculated media controls, incubating over multiple days, then comparing pH changes between inoculated and control conditions before sequencing the most acidic isolate. The AI agent instead performed repeated baseline pH measurements of cultures before incubation, then measured pH changes relative to each culture's own baseline rather than using external controls. Both approaches represent reasonable interpretations of the experimental goal, demonstrating how different interpretations can yield distinct but valid protocols.

*Case Study 3: Determine if individual zebrafish are nacre/nacre-type for breeding cross planning*

**AI agents miss domain-specific shortcuts that experts recognize.** In this experiment, GPT-5 and human experts pursued completely different strategies. The experiment contained a domain-specific shortcut: nacre-nacre mutants can be identified through visual inspection of tail cuttings for melanocyte presence. The state transition model correctly responded to visual inspection queries asking "Which tails are grey or black and which don't have any pigmented lines?", providing sufficient evidence to achieve the goal. However, no AI agents employed this approach, instead defaulting to DNA extraction and sequencing workflows typical of genotyping experiments. This case highlights the value of domain expertise and embodied reasoning in experimental design, where experts recognize efficient shortcuts that general reasoning approaches may overlook.

## C    MODEL PROMPTS

This section contains the complete prompts used for each component of our experimental planning system.

### C.1    AGENT PROMPT

The Agent prompt provides comprehensive context: the experimental goal, current materials with all properties and conditions, available actions with parameter specifications, experiment elapsed time, action history with outcomes, and recent classification results indicating progress toward the goal. This rich context enables the agent to make informed decisions about the experimental strategy.

Action selection follows a structured format. The agent must specify the action name from the available action set, exact material barcodes to use (preventing hallucination of materials), and parameters with appropriate values and units. For example, when performing flow cytometry, the agent specifies: action name (FlowCytometry), target materials (e.g., "HEK-293T cells in

PBS suspension (MAT-A1B2C3D4)"), and parameters (sample_volume=1mL, flow_rate=medium, event_count=10000, etc). The prompt structure includes:

```
CONTEXT: You are an agent in an experiment design simulation. Your goal
    is to reach a state of (materials, observations) that a competent
    scientist would interpret as sufficient evidence of a goal
    interpretation G. At each step, you will choose a set of materials,
    an action, and a set of parameters, and a simulator will execute the
    action and return the results in terms of new materials and
    observations.

TIMING: Different actions have different time costs. There is no penalty
    for short or long experiments, but some materials dynamically change
    over time and you should keep this in mind.

DYNAMIC/STATIC MATERIALS: Some materials in the list are dynamic,
    meaning they can change over time (e.g. grow, internalize, produce
    products, degrade, etc.). You should keep this in mind when
    designing your experiment.

QUITTING: If you do not believe you can achieve the goal with the
    current materials and observations, you can choose to quit the
    experiment with the Quit action.

FINAL SUBMISSION: If you believe you have achieved the goal, you can
    choose to submit the materials and observations for review with the
    SubmitMaterialsObservationsForReview action.

ACTION HISTORY: You will be given a history of actions and their results
    taken in the experiment so far.

RESPONSE FORMAT:
Respond with the action name, specific materials to use, and structured
    parameters in this format:
ACTION: ExperimentActionName
MATERIALS: Copy the exact material barcodes from the Current Materials
    list (including the hyphen: MAT-A1B2C3D4)
PARAMETERS: Select specific values from the parameter options shown for
    the chosen action

Although you are planning multiple steps ahead, you should only choose
    one action at a time.

Requirements:
- MATERIALS must be EXACT barcodes copied from the Current Materials
    list (format: MAT-A1B2C3D4)
- Copy the barcodes exactly as shown, including the hyphen after MAT
- List all materials that will be used, consumed, or modified by this
    action (whether directly or in the parameters)
- PARAMETERS must use the exact parameter names and values from the
    options shown
- All required parameters must be specified
- For multiselect parameters, specify multiple values separated by commas
- For material_selector parameters, specify the exact material barcode
    from the Current Materials list
- For multi_material_selector parameters, specify multiple material
    barcodes from the Current Materials list (comma-separated)
- For range parameters, specify a numeric value within the min-max range
    (e.g., wait_time=3600 for 1 hour)
- For range_with_units parameters, specify number + unit (e.g.,
    final_volume=500L, temperature=37C, wait_time=15min)
- For boolean parameters, specify "true" or "false"
- For object parameters, specify values in {key:value,key:value} format
```

```
Examples:
ACTION: ExperimentMix
MATERIALS: MAT-A1B2C3D4, MAT-E5F6G7H8
PARAMETERS: mixing_method=vortex, duration=1min, speed=medium

ACTION: ExperimentWait
MATERIALS: MAT-A1B2C3D4
PARAMETERS: wait_time=3600, conditions=37C_5%CO2

ACTION: ExperimentFlowCytometry
MATERIALS: MAT-A1B2C3D4
PARAMETERS: sample_volume=100L, flow_rate=medium, event_count=10000,
    threshold_value=250, detector_voltages={FSC:400,SSC:500,FL1:600},
    compensation_enabled=true

ACTION: SubmitMaterialsObservationsForReview
MATERIALS:
PARAMETERS:

ACTION: Quit
MATERIALS:
PARAMETERS:
```

## C.2 Transition simulator prompt

The transition simulator predicts experimental outcomes given state-action pairs:

```
TRANSITION_SYSTEM_PROMPT = """You are a scientific experiment simulator.
    Your task is to simulate the state transition from a current
    experimental state to the resulting materials and observations after
    performing an action on the current materials.

CRITICAL CONSTRAINTS:
1. You can ONLY work with the materials listed in the user message AND
    the materials explicitly mentioned in the action parameters -
    absolutely NO creating new reagents, samples, or materials
2. New materials can only result from transforming/combining the
    existing materials
3. You cannot assume the presence of any reagents, samples, materials,
    or equipment not explicitly listed (unless explicitly mentioned in
    the action parameters)

Examples:
- If action mentions using reagent X but reagent X is not in current
    materials  FAIL
- If action is "mix solutions A and B" and both are present  SUCCESS
    with realistic products
- If an action has parameter with an explicit material name like {...
    "stain": "DAPI" ...}, the action can use DAPI to stain the samples
    in the simulation because DAPI is explicitly mentioned in the action
    parameters, even if DAPI is not in current materials
- If an action has parameter with an explicit material barcode like {...
    "stain": "MAT-123456" ...} and MAT-123456 not in current materials
    FAIL.

Outcomes should be scientifically accurate. Consider:
- Chemical reactions and physical processes
- Conservation laws (conservation of mass, energy, etc.)
- Realistic experimental outcomes
- Time-related processes (growth, decay, transformation, etc.)
```

```
702    - Observational limits (observation should be constrained to exactly
703      what could be inferred from an action by a human or AI with
704      human-like sensory capabilities.)
705    - Material balance (what goes in must come out)
706
707    Respond in this JSON format:
708    {
709      "new_materials": [
710        {"name": "material_name", "properties": {"key": "value"},
711        "available_for_use":
712        "unlimited|available_for_further_use|not_available_for_further_use",
713        "material_type": "static|dynamic", "environmental_condition":
714        "room_temp|incubating_conditions|freezer|etc."}
715      ],
716      "new_observations": [
717        {"description": "detailed_observation_description",
718        "measurement_type": "visual|quantitative|qualitative|instrumental",
719        "value": "specific_measured_or_observed_value", "timestamp":
720        "optional_timestamp"}
721      ],
722      "reasoning": "Brief explanation of the predicted outcome",
723      "success": true/false---should be false if the action cannot be
724        performed based on missing materials or something physically
725        preventing initiating the action, true if the action can be
726        performed,
727      "failure_reason": "explain why action cannot be performed (what
728        materials are missing, what's preventing the action from being
729        performed) if success is false"
730    }
```

Important for materials:
- The "properties" of new materials should either appropriately inherit
  or update all the properties of the materials they are derived from,
  as well as any new properties that are not present in the materials
  they are derived from.
- Always be clear when you want to specify media/sample volume vs
  container volume, as this can mess things up.
- For each new material, you should include the following fields:
  - available_for_use:
    available_for_further_use|not_available_for_further_use.
    "available_for_further_use" means that the material can be used in
    future actions, "not_available_for_further_use" means that the
    material is no longer available for use (because it has been
    consumed, transformed into a new materials, etc.).
  - material_type: static|dynamic. "static" means that the material is
    not subject to significant change over time (e.g. properly stored
    reagents, buffers, equipment), "dynamic" means that the material is
    subject to significant change over time (e.g. cells, reactions,
    biological samples).
  - environmental_condition:
    room_temp|4C|-20C|-80C|-196C|incubating_conditions|cell_culture_conditions|ice|dry_ice.
    This is the environmental condition the material is in after the
    action is performed. For 'incubating_conditions', the specific
    temperature should be evident in the material properties (e.g.,
    temperature: 37C, 70C, etc.). For 'cell_culture_conditions', the
    specific culture parameters should be evident in the material
    properties (e.g., temperature: 37C, CO2: 5%, humidity: 95%, etc.).

Important for observations:
- description: Detailed description of what was observed
- measurement_type: One of "visual", "quantitative", "qualitative",
  "instrumental", "microscopy", "flow_cytometry", etc.
- value: Specific value, measurement, or qualitative assessment
- timestamp: Optional, can be step number or time

```
756   - Observations should only detail facts **directly** accessible from the
757       five senses and nothing more. For instance, visualizing a sample,
758       readable as a measurement from a device, smelling an odor, feeling a
759       texture or temperature, sensing heat or cold, hearing a sound, etc.
760       The observation model should assume that the agent doesn't actually
761       have access to the underlying material state, and therefore should
762       only state things that they could observed from interacting with the
763       sample(s) and and results of measurement(s).
764   Important for transition state fidelity:
765   - Do not mention the barcodes in the new materials or new observations,
766       as this can mess up the transition system.
767   - Observational and material properties should derive exclusively from
768       the limits of the action and the materials in the current state. For
769       example, after properly culturing HEK cells with AAV virus producing
770       genes, a plausible inference is that there are AAV particles in the
771       medium surrounding media, but one could not infer that these
772       particles are visible (because they are too small for the eye to
773       see) or infer what the titer is (because it is not measured).
774   - Ensure that observations and materials are mutually consistent.
775   - Finally, it is the job of the state management system to provide
776       sufficient information to model the state transitions. It is not
777       your job to add detail to the materials if they are critically
778       unclear. For example, if you need to transfer some amount of volume
779       from a material but the original volume of that material is not
780       explicit, then you do not need to try to infer a made-up value;
781       instead, you should return an error.
782   - You do not need to include input materials that have
783       "AVAILABLE_FOR_USE: unlimited" in the outcomes of the transitions,
784       as these are unmodifiable and persistent."""

      TRANSITION_USER_MESSAGE = """Current Materials:
      {materials}

      Current Observations:
      {observations}

      Action to Perform:
      {action}"""

      EXAMPLE_TRANSITIONS = [
          {
              "materials": [{"name": "sodium_chloride_solution", "properties":
          {"concentration": "0.1M", "volume": "100mL"}}],
              "observations": [{"description": "Clear colorless solution",
          "measurement_type": "visual", "value": "transparent"}],
              "action": {"name": "add_reagent", "parameters": {"reagent":
          "silver_nitrate", "amount": "10mL", "concentration": "0.1M"}},
              "outcome": {
                  "new_materials": [
                      {"name": "silver_chloride_precipitate", "properties":
          {"mass": "1.43g", "color": "white"}, "available_for_use":
          "available_for_further_use", "material_type": "dynamic",
          "environmental_condition": "room_temp"},
                      {"name": "sodium_nitrate_solution", "properties":
          {"concentration": "dilute", "volume": "110mL"}, "available_for_use":
          "available_for_further_use", "material_type": "dyanmic",
          "environmental_condition": "room_temp"}
                  ],
                  "new_observations": [
                      {"description": "White precipitate formed",
          "measurement_type": "visual", "value": "cloudy_white"},
                      {"description": "Precipitate mass", "measurement_type":
          "gravimetric", "value": "1.43g"}
```

```
810                    ]
811                }
812            }
813    ]
814

815
       INCORRECT_TRANSITION_PROMPT = """You are a scientific experiment
816        simulator. Given a current experimental state and a protocol action,
817        simulate the state transition to the resulting materials and
818        observations.
819
       Current Materials:
820    {materials}
821

822    Current Observations:
823    {observations}
824

825    Action to Perform:
       {action}
826

827    CRITICAL CONSTRAINTS:
828    1. You can ONLY work with the materials listed above AND the materials
829        explicitly mentioned in the action parameters- absolutely NO
830        creating new reagents, samples, or materials
       2. New materials can only result from transforming/combining the
831        existing materials
832    3. You cannot assume the presence of any reagents, samples, materials,
833        or equipment not explicitly listed (unless explicitly mentioned in
834        the action parameters)
835
       Examples:
836    - If action mentions using reagent X but reagent X is not in current
837        materials  FAIL
838    - If action is "mix solutions A and B" and both are present  SUCCESS
839        with realistic products
840
       Outcomes should be scientifically accurate. Consider:
841    - Chemical reactions and physical processes
842    - Conservation laws (conservation of mass, energy, etc.)
843    - Realistic experimental outcomes
844    - Time-related processes (growth, decay, transformation, etc.)
       - Observational limits (observation should be constrained to exactly
845        what could be inferred from an action by a human or AI with
846        human-like sensory capabilities.)
847    - Material balance (what goes in must come out)
848

849
       Respond in this JSON format:
850    {{
851      "new_materials": [
852        {{"name": "material_name", "properties": {{"key": "value"}},
853        "available_for_use":
854        "unlimited|available_for_further_use|not_available_for_further_use",
855        "material_type": "static|dynamic", "environmental_condition":
856        "room_temp|incubating_conditions|freezer|etc."}}
       ],
857      "new_observations": [
858        {{"description": "detailed_observation_description",
859        "measurement_type": "visual|quantitative|qualitative|instrumental",
860        "value": "specific_measured_or_observed_value", "timestamp":
861        "optional_timestamp"}}
       ],
862      "reasoning": "Brief explanation of the predicted outcome",
863
```

```
    "success": true/false---should be false if the action cannot be
      performed based on missing materials or physical constraints on
      initiating the action, true if the action can be performed,
    "failure_reason": "explain why action cannot be performed (what
      materials are missing, what physical constraints are preventing the
      action from being performed) if success is false"
}}

Important for materials:
- The "properties" of new materials should either appropriately inherit
    or update all the properties of the materials they are derived from,
    as well as any new properties that are not present in the materials
    they are derived from.
- Always be clear when you want to specify media/sample volume vs
    container volume, as this can mess things up.
- For each new material, you should include the following fields:
    - available_for_use:
      available_for_further_use|not_available_for_further_use.
      "available_for_further_use" means that the material can be used in
      future actions, "not_available_for_further_use" means that the
      material is no longer available for use (because it has been
      consumed, transformed into a new materials, etc.).
    - material_type: static|dynamic. "static" means that the material is
      not subject to significant change over time (e.g. properly stored
      reagents, buffers, equipment), "dynamic" means that the material is
      subject to significant change over time (e.g. cells, reactions,
      biological samples).
    - environmental_condition:
      room_temp|4C|-20C|-80C|-196C|incubating_conditions|cell_culture_conditions|ice|dry_ice.
      This is the environmental condition the material is in after the
      action is performed. For 'incubating_conditions', the specific
      temperature should be evident in the material properties (e.g.,
      temperature: 37C, 70C, etc.). For 'cell_culture_conditions', the
      specific culture parameters should be evident in the material
      properties (e.g., temperature: 37C, CO2: 5%, humidity: 95%, etc.).

Important for observations:
- description: Detailed description of what was observed
- measurement_type: One of "visual", "quantitative", "qualitative",
    "instrumental", "microscopy", "flow_cytometry", etc.
- value: Specific value, measurement, or qualitative assessment
- timestamp: Optional, can be step number or time
- Observations should only detail facts **directly** accessible from the
    five senses and nothing more. For instance, visualizing a sample,
    readable as a measurement from a device, smelling an odor, feeling a
    texture or temperature, sensing heat or cold, hearing a sound, etc.
    The observation model should assume that the agent doesn't actually
    have access to the underlying material state, and therefore should
    only state things that they could observed from interacting with the
    sample(s) and and results of measurement(s).

Important for transition state fidelity:
- Do not mention the barcodes in the new materials or new observations,
    as this can mess up the transition system.
- Observational and material properties should derive exclusively from
    the limits of the action and the materials in the current state. For
    example, after properly culturing HEK cells with AAV virus producing
    genes, a plausible inference is that there are AAV particles in the
    medium surrounding media, but one could not infer that these
    particles are visible (because they are too small for the eye to
    see) or infer what the titer is (because it is not measured).
- Ensure that observations and materials are mutually consistent.
- Finally, it is the job of the state management system to provide
    sufficient information to model the state transitions. It is not
    your job to add detail to the materials if they are critically
```

```
    unclear. For example, if you need to transfer some amount of volume
    from a material but the original volume of that material is not
    explicit, then you do not need to try to infer a made-up value;
    instead, you should return an error.
- You do not need to include input materials that have
    "AVAILABLE_FOR_USE: unlimited" in the outcomes of the transitions,
    as these are unmodifiable and persistent.

***THIS IS THE PROMPT GIVEN TO AN ACTUAL LLM SIMULATOR YOU ARE
    IMITATING, BUT YOUR JOB IS TO GENERATE TRANSITIONS THAT ARE SUBTLELY
    FALSE/NONPLAUSIBLE. THIS WILL BE USED TO TEST HUMANS ON IF THEY CAN
    TELL THE DIFFERENCE. BE SNEAKY IN HOW THINGS ARE WRONG. IN
    "reasoning" AFTER SAYING THE MISLEADING REASONING, WRITE "SECRET:
    <why the transition is actually wrong>".***"""

OBVIOUSLY_INCORRECT_TRANSITION_PROMPT = """You are a scientific
    experiment simulator. Given a current experimental state and a
    protocol action, simulate the state transition to the resulting
    materials and observations.

Current Materials:
{materials}

Current Observations:
{observations}

Action to Perform:
{action}

CRITICAL CONSTRAINTS:
1. You can ONLY work with the materials listed above AND the materials
    explicitly mentioned in the action parameters- absolutely NO
    creating new reagents, samples, or materials
2. New materials can only result from transforming/combining the
    existing materials
3. You cannot assume the presence of any reagents, samples, materials,
    or equipment not explicitly listed (unless explicitly mentioned in
    the action parameters)

Examples:
- If action mentions using reagent X but reagent X is not in current
    materials  FAIL
- If action is "mix solutions A and B" and both are present  SUCCESS
    with realistic products

Outcomes should be scientifically accurate. Consider:
- Chemical reactions and physical processes
- Conservation laws (conservation of mass, energy, etc.)
- Realistic experimental outcomes
- Time-related processes (growth, decay, transformation, etc.)
- Observational limits (observation should be constrained to exactly
    what could be inferred from an action by a human or AI with
    human-like sensory capabilities.)
- Material balance (what goes in must come out)

Respond in this JSON format:
{{
  "new_materials": [
    {{"name": "material_name", "properties": {{"key": "value"}},
    "available_for_use":
    "unlimited|available_for_further_use|not_available_for_further_use",
    "material_type": "static|dynamic", "environmental_condition":
    "room_temp|incubating_conditions|freezer|etc."}}
  ],
```

```
972      "new_observations": [
973        {{"description": "detailed_observation_description",
974        "measurement_type": "visual|quantitative|qualitative|instrumental",
975        "value": "specific_measured_or_observed_value", "timestamp":
976        "optional_timestamp"}}
977      ],
978      "reasoning": "Brief explanation of the predicted outcome",
979      "success": true/false---should be false if the action cannot be
980        performed based on missing materials or physical constraints on
981        initiating the action, true if the action can be performed,
982      "failure_reason": "explain why action cannot be performed (what
983        materials are missing, what physical constraints are preventing the
984        action from being performed) if success is false"
       }}

       Important for materials:
       - The "properties" of new materials should either appropriately inherit
           or update all the properties of the materials they are derived from,
           as well as any new properties that are not present in the materials
           they are derived from.
       - Always be clear when you want to specify media/sample volume vs
           container volume, as this can mess things up.
       - For each new material, you should include the following fields:
         - available_for_use:
           available_for_further_use|not_available_for_further_use.
           "available_for_further_use" means that the material can be used in
           future actions, "not_available_for_further_use" means that the
           material is no longer available for use (because it has been
           consumed, transformed into a new materials, etc.).
         - material_type: static|dynamic. "static" means that the material is
           not subject to significant change over time (e.g. properly stored
           reagents, buffers, equipment), "dynamic" means that the material is
           subject to significant change over time (e.g. cells, reactions,
           biological samples).
         - environmental_condition:
           room_temp|4C|-20C|-80C|-196C|incubating_conditions|cell_culture_conditions|ice|dry_ice.
           This is the environmental condition the material is in after the
           action is performed. For 'incubating_conditions', the specific
           temperature should be evident in the material properties (e.g.,
           temperature: 37C, 70C, etc.). For 'cell_culture_conditions', the
           specific culture parameters should be evident in the material
           properties (e.g., temperature: 37C, CO2: 5%, humidity: 95%, etc.).

       Important for observations:
       - description: Detailed description of what was observed
       - measurement_type: One of "visual", "quantitative", "qualitative",
           "instrumental", "microscopy", "flow_cytometry", etc.
       - value: Specific value, measurement, or qualitative assessment
       - timestamp: Optional, can be step number or time
       - Observations should only detail facts **directly** accessible from the
           five senses and nothing more. For instance, visualizing a sample,
           readable as a measurement from a device, smelling an odor, feeling a
           texture or temperature, sensing heat or cold, hearing a sound, etc.
           The observation model should assume that the agent doesn't actually
           have access to the underlying material state, and therefore should
           only state things that they could observed from interacting with the
           sample(s) and and results of measurement(s).

       Important for transition state fidelity:
       - Do not mention the barcodes in the new materials or new observations,
           as this can mess up the transition system.
       - Observational and material properties should derive exclusively from
           the limits of the action and the materials in the current state. For
           example, after properly culturing HEK cells with AAV virus producing
           genes, a plausible inference is that there are AAV particles in the
```

```
1026     medium surrounding media, but one could not infer that these
1027     particles are visible (because they are too small for the eye to
1028     see) or infer what the titer is (because it is not measured).
1029   - Ensure that observations and materials are mutually consistent.
1030   - Finally, it is the job of the state management system to provide
1031     sufficient information to model the state transitions. It is not
         your job to add detail to the materials if they are critically
1032     unclear. For example, if you need to transfer some amount of volume
1033     from a material but the original volume of that material is not
1034     explicit, then you do not need to try to infer a made-up value;
1035     instead, you should return an error.
1036   - You do not need to include input materials that have
1037     "AVAILABLE_FOR_USE: unlimited" in the outcomes of the transitions,
         as these are unmodifiable and persistent.
1038
1039   ***THIS IS THE PROMPT GIVEN TO AN ACTUAL LLM SIMULATOR YOU ARE
1040       IMITATING, BUT YOUR JOB IS TO GENERATE TRANSITIONS THAT ARE
1041       OBVIOUSLY FALSE/NONPLAUSIBLE IN SCIENTIFICALLY SOPHISTICATED WAYS.
1042       THIS WILL BE USED TO TEST HUMANS ON BASIC SCIENTIFIC REASONING. MAKE
            OBVIOUS VIOLATIONS OF CONSERVATION LAWS, THERMODYNAMICS, KINETICS,
1043       OR PHYSICAL CONSTRAINTS, BUT USE PROPER SCIENTIFIC TERMINOLOGY. IN
1044       "reasoning" AFTER SAYING THE MISLEADING REASONING, WRITE "SECRET:
1045       <why the transition violates basic scientific principles>".***"""
1046
       # System prompt for obviously incorrect transitions (for testing
1047       purposes)
1048   OBVIOUSLY_INCORRECT_TRANSITION_SYSTEM_PROMPT = """You are a scientific
1049       experiment simulator that generates obviously incorrect transitions
1050       for testing purposes. Your task is to simulate state transitions
1051       that contain obvious violations of fundamental scientific principles
            while using proper scientific terminology.
1052
1053   CRITICAL CONSTRAINTS:
1054   1. You can ONLY work with the materials listed in the user message –
1055      absolutely NO creating new reagents, samples, or materials
1056   2. New materials can only result from transforming/combining the
1057      existing materials
       3. You cannot assume the presence of any reagents, samples, materials,
1058      or equipment not explicitly listed
1059
1060   Your goal is to generate transitions that are OBVIOUSLY
1061       FALSE/NONPLAUSIBLE in scientifically sophisticated ways. Make clear
            violations of:
1062   - Conservation laws (mass, energy, momentum)
1063   - Thermodynamic principles (entropy, spontaneity, equilibrium)
1064   - Kinetic constraints (reaction rates, time scales)
1065   - Physical limitations (solubility, phase behavior, molecular size)
1066
       But maintain scientific vocabulary and structure to test basic
1067       scientific reasoning skills.
1068
1069   Respond in this JSON format:
1070   {
1071     "new_materials": [
         {"name": "material_name", "properties": {"key": "value"},
1072     "available_for_use":
1073     "unlimited|available_for_further_use|not_available_for_further_use",
1074     "material_type": "static|dynamic", "environmental_condition":
         "room_temp|incubating_conditions|freezer|etc."}
1075     ],
1076     "new_observations": [
1077       {"description": "detailed_observation_description",
1078       "measurement_type": "visual|quantitative|qualitative|instrumental",
1079       "value": "specific_measured_or_observed_value", "timestamp":
         "optional_timestamp"}
```

```
1080      ],
1081      "reasoning": "Brief explanation of the predicted outcome followed by
1082        SECRET: <explanation of why the transition violates basic scientific
1083        principles>",
1084      "success": true/false,
1085      "failure_reason": "explain why action failed if success is false"
1086    }
1087    Important: After providing the misleading reasoning, write "SECRET:
1088        <explanation of why the transition violates fundamental scientific
1089        laws>" in the reasoning field."""
1090
1091    INCORRECT_TRANSITION_PROMPT = """
1092    You are a scientific experiment simulator. Given a current
1093    experimental state and a protocol action, predict the resulting
1094    materials and observations.
1095
1096    Current Materials:
1097    {materials}
1098
1099    Current Observations:
1100    {observations}
1101
1102    Action to Perform:
1103    {action}
1104
1105    CRITICAL CONSTRAINTS:
1106    1. You can ONLY work with the materials listed above –
1107    absolutely NO creating new reagents, samples, or materials
1108    2. If the action requires ANY material not in the current
1109    materials list, it MUST fail
1110    3. New materials can only result from transforming/combining
1111    the existing materials
1112    4. You cannot assume the presence of any reagents, samples,
1113    materials, or equipment not explicitly listed
1114
1115    Examples:
1116    - If action mentions using reagent X but reagent X is not in
1117    current materials  FAIL
1118    - If action is "mix solutions A and B" and both are present
1119    SUCCESS with realistic products
1120
1121    Outcomes should be scientifically plausible. Consider:
1122    - Chemical reactions and physical processes
1123    - Conservation of mass and energy
1124    - Realistic experimental outcomes
1125    - Material balance (what goes in must come out)
1126
1127    Respond in this JSON format:
1128    {{
1129      "new_materials": [
1130        {{"name": "material_name", "properties": {{"key":
1131        "value"}}, "supply": "unlimited|consumable|consumed",
1132        "material_type": "static|dynamic",
1133        "environmental_condition":
          "room_temp|incubator|freezer|etc."}}
      ],
      "new_observations": [
        {{"description": "detailed_observation_description",
        "measurement_type":
        "visual|quantitative|qualitative|instrumental", "value":
        "specific_measured_or_observed_value", "timestamp":
        "optional_timestamp"}}
```

```
1134       ],
1135       "reasoning": "Brief explanation of the predicted outcome",
1136       "success": true/false,
1137       "failure_reason": "explain why action failed if success is
1138       false"
1139     }}
1140
1141     Important for materials:
1142     - The "properties" of new materials should either approriately
         inherit or update all the properties of the materials they are
1143     derived from, as well as any new properties that are not
         present in the materials they are derived from.
1144     - Lastly, for each new material, you should include the
1145     following fields:
1146       - supply: consumable|consumed. "consumable" means that the
1147       material can be used in future actions, "consumed" means that
1148       the material is no longer available for use.
1149       - material_type: static|dynamic. "static" means that the
1150       material is not subject to significant change over time (e.g.
1151       properly stored reagents, buffers, equipment), "dynamic"
1152       means that the material is subject to significant change over
1153       time (e.g. cells, reactions, biological samples).
1154       - environmental_condition:
       room_temp|4C|-20C|-80C|-196C|37C|37C_5\%CO2|ice|dry_ice.
1155       This is the environmental condition of the material while the
         action is performed.
1156
1157     Important for observations:
1158     - description: Detailed description of what was observed
1159     - measurement_type: One of "visual", "quantitative",
         "qualitative", "instrumental", "microscopy", "flow_cytometry",
1160     etc.
1161     - value: Specific value, measurement, or qualitative assessment
1162     - timestamp: Optional, can be step number or time
1163
1164     ***THIS IS THE PROMPT GIVEN TO AN ACTUAL LLM SIMULATOR YOU ARE
         IMITATING, BUT YOUR JOB IS TO GENERATE TRANSITIONS THAT ARE
1165     SUBTLELY FALSE/NONPLAUSIBLE. THIS WILL BE USED TO TEST HUMANS
1166     ON IF THEY CAN TELL THE DIFFERENCE. BE SNEAKY IN HOW THINGS ARE
         WRONG. IN "reasoning" AFTER SAYING THE MISLEADING REASONING,
1167     WRITE "SECRET: <why the transition is actually wrong>".***
1168
```

## C.3 EVIDENCE CLASSIFIER PROMPT

The classifier evaluates whether the cumulative experimental observations constitute valid evidence for experimental goals. The classifier receives a comprehensive evidence summary containing the complete experimental history: all actions performed (with parameters and target materials), initial materials for each action, timing information, and all the resulting observations organized chronologically.

This evidence history is formatted as a structured text summary and evaluated against the experimental goal. The classification prompt instructs the model to consider scientific plausibility, logical consistency, appropriate level of inference, alternative explanations, evidence strength, and potential confounding factors across the entire experimental sequence. The classifier returns structured assessments including a binary verdict (VALID/INVALID) on goal achievement, scientific reasoning for the decision, confidence score (0.0-1.0), goal relevance assessment, and evidence strength rating.

```
CLASSIFICATION_PROMPT = """You are evaluating whether the experimental
    steps taken in a simulated wet lab environment are sufficient to
    reach an information state for some goal interpretation.

Simulation limits:
Because the experiment is simulated, it cannot supply exact numerical
    data (e.g. exact fluorescence brightness values, exact colony
```

```
counts, etc.). Therefore, the information state is defined by the
things which are derivable from the materials produced and the
observations produced. For instance, fluorescence intensity is
derivable from collected imaging files, explicit cell counts are
derivable from properly imaged propidium iodide staining, etc.
However, this would not be explicitly described in the experimental
rollout.

Appropriate inference:
You should judge the sufficiency of the information state based upon any
claim that should be inferable from the materials (in their
physical, embodied form) and the observations stated.

Given an actions, materials, and observations, and an interpretation,
determine if the simulated rollout provides the evidence to justify
the goal interpretation. If it does, consider the interpretation
valid. If it does not, consider the interpretation invalid.

Consider:
- Scientific plausibility
- Logical consistency
- What can be inferred from the materials and observations
- Embodied common sense
- Alternative explanations
- Strength of evidence
- Potential confounding factors

Experimental Steps: {observation}

Interpretation: {interpretation}

Respond in this format:
<verdict>VALID or INVALID</verdict>
<reasoning>
Explain why the interpretation is valid or invalid, including:
- What evidence supports or contradicts the interpretation
- Any scientific principles that apply
- Potential alternative explanations
- Strength of the logical connection
</reasoning>
<confidence>0.0 to 1.0</confidence>
<goal_relevance>directly supports goal | partially supports goal | does
    not support goal</goal_relevance>
<evidence_strength>strong | moderate | weak</evidence_strength>

Response:"""

SUMMARY_PROMPT = """You are a scientific summary writer. Your task is to
    summarize a given observation and interpretation into a concise
    summary.

Given an observation and an interpretation, summarize the observation
    and interpretation into a concise summary.

Observation: {observation}

Interpretation: {interpretation}
"""

CLASSIFICATION_EXAMPLES = [
    {
        "observation": "Solution turned blue when copper sulfate was
    added",
        "interpretation": "Copper ions are present in the solution",
```

```
        "label": "VALID"
    },
    {
        "observation": "Temperature increased by 5C during reaction",
        "interpretation": "The reaction is exothermic",
        "label": "VALID"
    },
    {
        "observation": "No precipitate formed when silver nitrate was
    added",
        "interpretation": "No chloride ions are present",
        "label": "VALID"
    },
    {
        "observation": "Solution turned red",
        "interpretation": "The molecule structure has been completely
    determined",
        "label": "INVALID"
    },
    {
        "observation": "pH decreased from 7 to 4",
        "interpretation": "A nuclear reaction has occurred",
        "label": "INVALID"
    }
]
```

# A    MODEL CONFIGURATIONS

The model configurations used in the Agent and environment models:

## A.1    AGENT MODEL CONFIGURATIONS

### A.1.1    GPT-5

```
provider_type: openai
model_name: gpt-5
temperature: 0.7
max_tokens: 16000
```

### A.1.2    CLAUDE 4 SONNET

```
provider_type: claude
model_name: claude-4-sonnet-20250514
temperature: 0.7
max_tokens: 4096
thinking_budget: 2048
```

### A.1.3    GPT-4O

```
provider_type: openai
model_name: gpt-4o
temperature: 0.7
max_tokens: 4096
```

### A.1.4    GEMINI 2.5 FLASH

```
provider_type: gemini
model_name: gemini-2.5-flash
temperature: 0.7
max_tokens: 16000
```

### A.1.5 QWEN3-235B (THINKING)

```
provider_type: openrouter
model_name: qwen/qwen3-235b-a22b-thinking-2507
temperature: 0.7
max_tokens: 16000
```

### A.1.6 DEEPSEEK-R1

```
provider_type: openrouter
model_name: deepseek/deepseek-r1-0528
temperature: 0.7
max_tokens: 16000
```

### A.1.7 STATE TRANSITION MODEL (CLAUDE 4 SONNET)

```
provider_type: claude
model_name: claude-4-sonnet-20250514
temperature: 0.7
max_tokens: 16384
thinking_budget: 4096
error_type: 0  # 0=correct, 1=obviously wrong, 2=subtly wrong
```

### A.1.8 EVIDENCE CLASSIFIER (GPT-5)

```
provider_type: openai
model_name: gpt-5
temperature: 0.7
max_tokens: 16000
thinking_budget: None
```

## B COMPLETE EXPERIMENTAL GOALS DATASET

This section contains the complete list of 18 experimental goals used in our evaluation, spanning chemistry, biology, and bioengineering domains.

### B.1 CHEMISTRY

- **Caffeine HPLC Analysis**: Information required to determine the caffeine concentration in commercial coffee, tea, and energy drink samples (100 max steps)

- **Chloride Gravimetric Analysis**: Information required to determine the weight percentage of chloride in the unknown sample based on AgCl precipitate formation (100 max steps)

- **Ferrocene Cyclic Voltammetry**: Information required to determine the formal potential and reversibility of ferrocene oxidation in acetonitrile electrolyte (100 max steps)

### B.2 MICROBE CARBON CAPTURE

- **Acidifying Microbes**: Information required to characterize which acidic soil isolates from soil samples achieve the greatest pH reduction when grown on glucose substrates (100 max steps)

- **Biofilm-Forming Microbes**: Information required to determine which rock surface isolates form the most dense biofilms on basalt chip surfaces as measured by crystal violet staining (100 max steps)

- **Weathering Enhancement Quantification**: Information required to determine the total dissolved solute production from fluorapatite by microbial consortia compared to individual species over 14 days (50 max steps)

### B.3 Neural Biosensors

- **ASAP3 Voltage Biosensor**: Information required to prepare different concentrations (5 mM, 20 mM, 40 mM, 60 mM) of extracellular potassium solutions (HBSS-K5, HBSS-K20, HBSS-K40, HBSS-K60) with proper ionic composition, pH adjustment, and sterile filtration for voltage biosensor experiments (100 max steps)

- **ChR2-GCaMP Fusion**: Information required to characterize the correlation between 488nm stimulation power (in mW/mm$^2$) and ChR2 evoked calcium response amplitude in E18 rat primary cortical neurons (DIV 10-15) (100 max steps)

- **Dual FRET Biosensor**: Information required to determine simultaneous calcium activity and cAMP FRET ratio changes in HEK293T cells following $10\mu$M ionomycin and $50\mu$M forskolin treatment (100 max steps)

- **GCaMP7 Viral Packaging**: Information required to determine calcium response dynamics ($\Delta$F/F amplitude and kinetics) in E18 rat primary cortical neurons (DIV 7-10) following 50mM KCl depolarization (100 max steps)

- **iGluSnFR3 Glutamate Biosensor**: Information required to determine glutamate detection sensitivity and dynamic range in cultured E18 Sprague-Dawley rat hippocampal neurons expressing iGluSnFR3 across L-glutamate concentrations from $1\mu$M to 1mM (100 max steps)

### B.4 Minicell

- **Minicell Uptake**: Information required to characterize the effect of minicell:cell ratio on uptake (50 max steps)

### B.5 Genetics, lentivirus and Zebrafish

- **GFP Colony Screening**: Information required to know which of the 20 colonies contain correctly oriented and in-frame GFP inserts without mutations (100 max steps)

- **Lentivirus MOI Optimization**: Information required to calculate the titer of lentivirus (100 max steps)

- **P2A Cleavage Efficiency**: Information required to know the percentage efficiency of P2A-mediated cleavage between GFP and mCherry proteins in the fusion construct (50 max steps)

- **Puromycin Concentration Optimization**: Information to determine the minimum puromycin concentration that achieves ¿90% GFP-positive cells (50 max steps)

- **Zebrafish Genotyping**: Information required to know if individual zebrafish for breeding cross planning are nacre/nacre-type or not (50 max steps)

- **Zebrafish Progeny Genotyping**: Information required to know the genotype (WT/WT, WT/nacre, or nacre/nacre) of each individual progeny fish from the cross (50 max steps)

## C  Action Space Constraints

The available action sets are domain-specific and designed to reflect realistic laboratory capabilities:

- **SerialDilute**: Performs a series of dilutions iteratively by mixing samples with diluents and transferring to another container of the diluent. Expected input materials: One sample, one diluent, container(s) to hold the dilutions. If action is successful, expected output materials: separate materials for each dilution, residual sample (if any), residual diluent (if any).

- **Aliquot**: Generates a series new samples by drawing from a source sample. Expected input materials: One source sample, container(s) to hold the aliquots. If action is successful, expected output materials: separate materials for each aliquot, residual sample (if any).

- **Transfer**: Moves an amount of sample from a specified source to one or more specified destination vessels. Expected input materials: One source material, one or more destination materials. If action is successful, expected output materials: residual source material (if any), an individual material for each destination material with the transferred amount.

- **Wash**: Aspirates current media, performs gentle washing with chosen media, and replaces with fresh media. Expected input materials: One or more materials to wash, one wash media, one replacement media. If action is successful, expected output materials: separate materials for each wash, residual wash media (if any), residual replacement media (if any).

- **DNASynthesis**: Performs solid-phase deoxyribonucleic acid oligonucleotide synthesis of the given sequence or set of sequences using phosphoramidite chemistry. Expected input materials: One or more sequences to be synthesized. If action is successful, expected output materials: separate materials for each synthesized sequence.

- **RNASynthesis**: Performs solid-phase ribonucleic acid oligonucleotide synthesis of the given sequence or set of sequences using phosphoramidite chemistry. Expected input materials: One or more sequences to be synthesized. If action is successful, expected output materials: separate materials for each synthesized sequence.

- **PNASynthesis**: Performs solid-phase peptide synthesis of a given Peptide Nucleic Acid (PNA) sequence or set of sequences using Boc or Fmoc strategies. Expected input materials: One or more sequences to be synthesized. If action is successful, expected output materials: separate materials for each synthesized sequence.

- **Thermocycler**: Uses a thermocycler on the provided samples. Assume that all materials are transformed separately. Expected input materials: One or more materials to be thermocycled. If action is successful, expected output materials: separate materials for each thermocycled material.

- **PeptideSynthesis**: Performs classical solution phase synthesis of amino acids. Expected input materials: One or more sequences to be synthesized. If action is successful, expected output materials: separate materials for each synthesized sequence.

- **CapillaryElectrophoresis**: Performs capillary electrophoresis to separate and analyze analyte molecules in the given samples based on their electrophoretic mobility through a capillary filled with buffer solution. Expected input materials: One or more samples to be analyzed. If action is successful, expected output materials: separate materials for each analyzed sample with separation and detection data.

- **SolidPhaseExtraction**: Performs Solid Phase Extraction (SPE) to purify analyte molecules in the given samples by adsorbing analytes to a solid-phase resin, washing the resin with wash buffer to remove impurities, and then eluting the analyte from the solid phase using an elution buffer. Expected input materials: One or more samples to be purified, wash buffer, elution buffer. If action is successful, expected output materials: separate materials for each extracted sample, residual wash buffer (if any), residual elution buffer (if any).

- **HPLC**: Performs High Pressure Liquid Chromatography (HPLC) to separate analyte molecules in the given samples on the basis of their relative affinity to a mobile phase and a solid phase by flowing mobile phase through columns at high pressures. Expected input materials: One or more samples to be analyzed. If action is successful, expected output materials: separate materials for each analyzed sample, separate detection data for each analyzed sample.

- **AgaroseGelElectrophoresis**: Performs agarose gel electrophoresis to separate analyte molecules in a given sample on the basis of their electrophoretic mobility though an agarose gel. Expected input materials: One or more samples to be analyzed. If action is successful, expected output materials: the resulting gel with separated analyte molecules and an image of the gel from a blue light transilluminator.

- **PAGE**: Performs Polyacrylamide Gel Electrophoresis (PAGE) to separate analyte molecules in a given sample on the basis of their electrophoretic mobility though a polyacrylamide slab gel. Expected input materials: One or more samples to be analyzed. If action is successful, expected output materials: the resulting gel with separated analyte molecules and an image of the gel in a well lit room.

- **CapillaryWestern**: Performs a capillary-based analogous to the traditional Western blot to detect the presence of a specific protein in a given sample. Expected input materials: One or more samples to be analyzed. If action is successful, expected output materials: separate materials for each analyzed sample with protein detection data.

- **Dialysis**: Performs separation to remove small unwanted compounds by diffusion through a semipermeable membrane. Expected input materials: One or more samples to be dialyzed separately, dialysis buffer. If action is successful, expected output materials: separate materials for each dialyzed sample, residual dialysis buffer (if any).

- **MassSpectrometry**: Ionizes the given samples in order to measure the mass-to-charge ratio of the molecules in the samples. Expected input materials: One or more samples to be analyzed. If action is successful, expected output materials: separate materials for each analyzed sample after the measurement, separate mass spectrometry data for each analyzed sample.

- **TotalProteinQuantification**: Performs an absorbance- or fluorescence-based assay to determine the total protein concentration of given input samples. Expected input materials: One or more samples to be quantified. If action is successful, expected output materials: separate materials for each analyzed sample after the assay, separate protein concentration data for each analyzed sample.

- **qPCR**: Performs a quantitative polymerase chain reaction (qPCR) which uses a thermocycler to amplify a target sequence (or sequences if multiplexing) from the sample using a primer set, quantifying the amount of DNA or RNA throughout the using a fluorescent intercalating dye or fluorescently labeled probe. Expected input materials: One or more samples to be analyzed, primers, probe (if applicable). If action is successful, expected output materials: separate materials for each analyzed sample after amplification, separate amplification data for each analyzed sample, residual primers (if any), residual probe (if any).

- **BioLayerInterferometry**: Quantifies the magnitude and kinetics of an interaction between a surface immobilized species and a solution phase analyte sample. Expected input materials: One or more analyte samples to be analyzed. If action is successful, expected output materials: separate materials for each analyzed sample after the measurement, separate binding kinetics data for each analyzed sample.

- **CapillaryELISA**: Performs capillary Enzyme-Linked Immunosorbent Assay (ELISA) on the provided Samples for the detection of certain analytes. Expected input materials: One or more samples to be analyzed. If action is successful, expected output materials: separate materials for each analyzed sample after the assay, separate analyte detection data for each analyzed sample.

- **ELISA**: Performs a quantitative characterization of the specific antigen concentration in samples. Expected input materials: One or more samples to be analyzed. If action is successful, expected output materials: separate materials for each analyzed sample after the assay, separate antigen concentration data for each analyzed sample.

- **DNASequencing**: Identifies the order of nucleotides in a strand of DNA. Expected input materials: One or more DNA samples to be sequenced. If action is successful, expected output materials: separate materials for each sequenced sample after the sequencing, separate DNA sequence data for each sequenced sample.

- **NucleicAcidQuantification**: Use DNA spectrophotometry to determine the concentration and purity of DNA, RNA, or other nucleic acids in a given sample. Expected input materials: One or more nucleic acid samples to be quantified. If action is successful, expected output materials: separate materials for each analyzed sample after the measurement, separate nucleic acid concentration and purity data for each analyzed sample.

- **FlowCytometry**: Performs flow cytometry to analyze the characteristics of individual cells or particles in a fluid by passing them through a laser beam, where their size, shape, and fluorescent labels are measured. Expected input materials: One or more cell or particle samples to be analyzed. If action is successful, expected output materials: separate materials for each analyzed sample after the measurement, separate flow cytometry data for each analyzed sample.

- **Dilute**: Adds a specified amount of solvent to specified samples. Expected input materials: One or more samples to be diluted, solvent. If action is successful, expected output materials: separate materials for each diluted sample, residual solvent (if any).

- **Incubate**: Heats and/or mixes the provided samples for a given amount of time at a given temperature, allowing for a follow up annealing time. Expected input materials: One or more samples to be incubated. If action is successful, expected output materials: separate materials for each incubated sample.

- **DryingOven**: Dry samples in a drying oven. Expected input materials: One or more samples to be dried. If action is successful, expected output materials: separate materials for each dried sample.

- **Mix**: Mixes and/or heats the provided samples for a given amount of time at a given rate and temperature. Expected input materials: One or more samples to be mixed. If action is successful, expected output materials: separate materials for each mixed sample.

- **Combine**: Combines the provided samples into a single sample. Expected input materials: Two or more samples to be combined. If action is successful, expected output materials: one combined material containing all input samples.

- **Centrifuge**: Spins down the provided samples for a given amount of time at a provided force or spin rate. Expected input materials: One or more samples to be centrifuged. If action is successful, expected output materials: separate materials for each centrifuged sample.

- **Degas**: Performs a degassing procedure on the given samples using a specified technique. Expected input materials: One or more samples to be degassed. If action is successful, expected output materials: separate materials for each degassed sample.

- **Filter**: Passes the provided samples through a given physical filter using a set of optional different methods. Expected input materials: One or more samples to be filtered. If action is successful, expected output materials: separate materials for each filtered sample, collected filtrate materials.

- **Autoclave**: Subjects the provided samples or containers to extreme heat and pressure in order to sterilize. Expected input materials: One or more samples or containers to be sterilized. If action is successful, expected output materials: separate materials for each sterilized sample or container.

- **Evaporate**: Evaporates solvent from a provided sample under high vacuum at a given temperature with centrifugation to prevent bumping. Expected input materials: One or more samples to have solvent evaporated. If action is successful, expected output materials: separate materials for each sample with reduced solvent volume.

- **Lyophilize**: Removes solvents from the provided samples via controlled freezing and sublimation under high vacuum. Expected input materials: One or more samples to be lyophilized. If action is successful, expected output materials: separate materials for each lyophilized sample.

- **Aspirate**: Removes the supernatant from a sample by aspiration. Expected input materials: One or more samples with supernatant to be removed. If action is successful, expected output materials: separate materials for each sample with supernatant removed, collected supernatant materials.

- **FillToVolume**: Adds sample to the a container until its volume reaches the desired value. Expected input materials: One or more samples to be filled to volume, solvent for volume adjustment. If action is successful, expected output materials: separate materials for each sample adjusted to target volume, residual solvent (if any).

- **AcousticLiquidHandling**: Transfers liquid samples with sound waves in nanoliter increments. Expected input materials: One or more source samples, destination containers. If action is successful, expected output materials: residual source materials (if any), separate materials for each destination with transferred amounts.

- **AdjustpH**: Adds acid or base titrant to change the pH of the given sample to the desired value. Expected input materials: One or more samples to have pH adjusted, acid or base titrant. If action is successful, expected output materials: separate materials for each pH-adjusted sample, residual titrant (if any).

- **Resuspend**: Dissolve the specified solid samples with some amount of solvent. Can apply to living and non-living solid samples. Expected input materials: One or more solid samples to be resuspended, solvent. If action is successful, expected output materials: separate materials for each resuspended sample, residual solvent (if any).

- **ApplyMagnet**: Isolates targets from specified sample via magnetic bead separation, which uses a magnetic field to separate superparamagnetic particles from suspensions. Expected input materials: One or more samples containing magnetic beads. If action is successful, expected output materials: separate materials for each sample with magnetically separated components.

- **Microwave**: Breaks down complex samples via microwave heating and (optional) acid/oxidizing agent to fully solubilize sample for subsequent operations, especially ICP-MS. Expected input materials: One or more samples to be microwaved. If action is successful, expected output materials: separate materials for each microwaved sample.

- **FlashFreeze**: Performs freezing of specified sample objects through immersion of the sample containers in liquid nitrogen. Expected input materials: One or more samples to be flash frozen. If action is successful, expected output materials: separate materials for each flash frozen sample.

- **Desiccate**: Dries out solid substances by absorbing water molecules from the samples through exposing them to a chemical desiccant in a bell jar desiccator under vacuum or non-vacuum conditions. Expected input materials: One or more samples to be desiccated. If action is successful, expected output materials: separate materials for each desiccated sample.

- **Grind**: Employs mechanical actions to break particles of solid samples into smaller powder particles, using a grinding apparatus. Expected input materials: One or more solid samples to be ground. If action is successful, expected output materials: separate materials for each ground sample.

- **AttemptDNAExtraction**: Series of steps to isolate DNA from a given sample. Also, commonly referred to as a mini-prep, midi-prep, etc. Expected input materials: One or more samples containing DNA to be extracted. If action is successful, expected output materials: separate materials for each extracted DNA sample.

- **CountLiquidParticles**: Measures the number of suspended particles in a liquid colloid or very fine suspension sample. Expected input materials: One or more liquid samples with suspended particles to be counted. If action is successful, expected output materials: separate materials for each analyzed sample after the measurement, separate particle count data for each analyzed sample.

- **CoulterCount**: Measures the number and size distribution of suspended particles (typically cells) in a liquid colloid or very fine suspension sample. Expected input materials: One or more samples with suspended particles to be counted. If action is successful, expected output materials: separate materials for each analyzed sample after the measurement, separate particle count and size distribution data for each analyzed sample.

- **MeasureOsmolality**: Measures the concentration of osmotically active species in a solution. Expected input materials: One or more samples to have osmolality measured. If action is successful, expected output materials: separate materials for each analyzed sample after the measurement, separate osmolality data for each analyzed sample.

- **MeasureConductivity**: Measures the electrical conductivity of a sample by immersion of a conductivity probe into the solution. Expected input materials: One or more samples to have conductivity measured. If action is successful, expected output materials: separate materials for each analyzed sample after the measurement, separate conductivity data for each analyzed sample.

- **MeasureContactAngle**: Measures the contact angle of a fiber sample with a wetting liquid using a force tensiometer. Expected input materials: One or more fiber samples, wetting liquid. If action is successful, expected output materials: separate materials for each analyzed sample after the measurement, separate contact angle data for each analyzed sample, residual wetting liquid (if any).

- **MeasureDensity**: Measures the density of the given samples using a fixed volume weight measurement or a density meter. Expected input materials: One or more samples to have density measured. If action is successful, expected output materials: separate materials for each analyzed sample after the measurement, separate density data for each analyzed sample.

- **MeasureDissolvedOxygen**: Measures the partial pressure of oxygen in a sample by applying a constant voltage in a probe confined by an oxygen permeable membrane to detect oxygen reduction as an electrical signal. Expected input materials: One or more samples to have dissolved oxygen measured. If action is successful, expected output materials: separate materials for each analyzed sample after the measurement, separate dissolved oxygen data for each analyzed sample.

- **MeasurepH**: Measures the pH of the given sample using electrical potential sensors. Expected input materials: One or more samples to have pH measured. If action is successful, expected output materials: separate materials for each analyzed sample after the measurement, separate pH data for each analyzed sample.

- **MeasureWeight**: Measures the weight of the given samples using an appropriately sized balance. Expected input materials: One or more samples to have weight measured. If action is successful, expected output materials: separate materials for each analyzed sample after the measurement, separate weight data for each analyzed sample.

- **MeasureVolume**: Measures the volume of the given samples using ultrasonic measurement of liquid surface distance and prior parametrization of the surface distance to volume in the samples container to determine sample volumes. Expected input materials: One or more samples to have volume measured. If action is successful, expected output materials: separate materials for each analyzed sample after the measurement, separate volume data for each analyzed sample.

- **MeasureCount**: Measures the number of tablets in a given tablet sample by determining the average weight of the tablets in the sample and the total mass of the given tablet sample. Expected input materials: One or more tablet samples to have count measured. If action is successful, expected output materials: separate materials for each analyzed sample after the measurement, separate tablet count data for each analyzed sample.

- **ImageSample**: Records an image of the given sample either from above or side on for larger transparent vessels. Expected input materials: One or more samples to be imaged. If action is successful, expected output materials: separate materials for each imaged sample after the imaging, separate image data for each imaged sample.

- **MeasureSurfaceTension**: Determines the surface tension of a sample by measuring the forces exerted on a small diameter rod as it is withdrawn from a sample. Expected input materials: One or more samples to have surface tension measured. If action is successful, expected output materials: separate materials for each analyzed sample after the measurement, separate surface tension data for each analyzed sample.

- **MeasureRefractiveIndex**: Measures the Refractive Index (RI) of the given sample with refractometer. Expected input materials: One or more samples to have refractive index measured. If action is successful, expected output materials: separate materials for each analyzed sample after the measurement, separate refractive index data for each analyzed sample.

- **CyclicVoltammetry**: Characterizes the reduction and oxidation processes of the given sample using Cyclic Voltammetry (CV). Expected input materials: One or more samples to be analyzed, reference electrode. If action is successful, expected output materials: separate materials for each analyzed sample after the measurement, separate cyclic voltammetry data for each analyzed sample.

- **PrepareReferenceElectrode**: Generates a reference electrode filled with a reference solution to be used in electrochemical measurements, including Cyclic Voltammetry measurements. Expected input materials: None. If action is successful, expected output materials: prepared reference electrode.

- **VisualInspection**: Monitors the insoluble particles in the given sample while its container is agitated. Expected input materials: One or more samples to be visually inspected. If ac-

tion is successful, expected output materials: separate materials for each inspected sample after the inspection, separate visual inspection data for each inspected sample.

- **MeasureViscosity**: Measures a fluid's viscosity defined as the resistance to deformation by assessing the flow rate of the sample when loaded into the viscometer chip. Expected input materials: One or more fluid samples to have viscosity measured. If action is successful, expected output materials: separate materials for each analyzed sample after the measurement, separate viscosity data for each analyzed sample.

- **DynamicFoamAnalysis**: Characterizes the foamability, stability, drainage process and structure of liquid-based foams by monitoring foam generation and decay of a sample. Expected input materials: One or more liquid samples to have foam properties analyzed. If action is successful, expected output materials: separate materials for each analyzed sample after the analysis, separate foam analysis data for each analyzed sample.

- **MeasureMeltingPoint**: Measures the melting points of the solid samples using a melting point apparatus that applies an increasing temperature gradient to melting point capillary tubes containing a small amount of the input samples. Expected input materials: One or more solid samples to have melting point measured. If action is successful, expected output materials: separate materials for each analyzed sample after the measurement, separate melting point data for each analyzed sample.

- **UseMicroscope**: Performs imaging on provided cellular samples using a bright-field microscope or a high content imager. Expected input materials: One or more cellular samples to be imaged. If action is successful, expected output materials: separate materials for each imaged sample after the imaging, separate microscopy data for each imaged sample.

- **ImageColonies**: Acquires bright-field, absorbance or fluorescence images of the provided samples containing microbial cells on a solid media plate using a colony handler. Expected input materials: One or more samples containing microbial colonies to be imaged. If action is successful, expected output materials: separate materials for each imaged sample after the imaging, separate colony image data for each imaged sample.

- **FreezeCells**: Lowers the temperature of cell samples under controlled conditions to prepare cells for long term cryopreservation. Expected input materials: One or more cell samples to be frozen. If action is successful, expected output materials: separate materials for each frozen sample.

- **QuantifyColonies**: Measures the microbial cell concentration in the provided samples. Expected input materials: One or more samples containing microbial colonies to be quantified. If action is successful, expected output materials: separate materials for each analyzed sample after the quantification, separate colony count data for each analyzed sample.

- **CellCultureIncubate**: Incubate samples in sterile, controlled conditions. Expected input materials: One or more cell culture samples to be incubated. If action is successful, expected output materials: separate materials for each incubated sample.

- **Wait**: Allows the system to idle for a specified amount of time to let materials evolve, react, or reach equilibrium. Expected input materials: One or more materials to be waited on. If action is successful, expected output materials: separate materials for each material after waiting period.

- **Storage**: Store materials under specified environmental conditions for preservation or appropriate handling. Expected input materials: One or more materials to be stored. If action is successful, expected output materials: separate materials for each stored material.

- **UsePhotospectrometer**: Measure absorbance, transmittance, or fluorescence using photospectrometer. Expected input materials: One or more samples to be analyzed. If action is successful, expected output materials: separate materials for each analyzed sample after the measurement, separate spectrophotometric data for each analyzed sample.

- **UsePlateReader**: Measure absorbance, fluorescence, or luminescence using microplate reader. Expected input materials: One or more samples in microplate format to be analyzed. If action is successful, expected output materials: separate materials for each analyzed sample after the measurement, separate plate reader data for each analyzed sample.

- **SubmitMaterialsObservationsForReview**: When you believe you have a set of materials and observations that meet the all of the requirements for the goal, submit them for final review.

- **Quit**: If you do not believe you can complete the task with the current materials and observations, quit the process.

- **ManifestMaterials**: Request and add new laboratory materials to the inventory using natural language descriptions. This action has no time cost and does not advance the step count.

- **SeparateWells**: Separates a multi-well plate material into individual well materials or groups based on treatment conditions. Importantly, this is not a laboratory action, but rather a simulation-specific "perception" action to help the simulation agent interact with its available materials by being able to use them as individual samples. The properties of the resulting materials should not change as a result of this action, and no time passes as a result of this action. Rather, they should be properly preserved and simply broken up into individual materials.

- **CellCounter**: Counts the number of cells in a given sample using automated cell counting methods such as hemocytometer, automated cell counter, or flow cytometry. Input material is consumed and is no longer available for use after this method.

- **Discard**: Discards unwanted materials or samples by disposing of them safely according to appropriate waste disposal protocols.

