# OpenReview forum: "WetBench: LLM-Based Simulation Environment to Evaluate Wet-Lab Experiment Planning and Design"
_ICLR.cc/2026/Conference — Submitted to ICLR 2026_

### Official Review · Reviewer_fuRW · 2025-10-26

**Soundness:** 2
**Presentation:** 2
**Contribution:** 2
**Rating:** 2
**Confidence:** 3

**Summary:**

### Overview
This paper introduces WetBench, a simulation environment intended to scalably evaluate the ability of large language models (LLMs) to plan and design wet-lab experiments. The authors propose to solve the problem of expense and safety in real-world lab execution by creating a fully LLM-based simulation. The environment consists of two core components: (1) a "state transition model," an LLM that takes the current state and an agent's action to predict a new state and textual observation, and (2) an "evidence classifier," another LLM that evaluates the history of observations to determine if the experimental goal has been met. The authors use this environment to benchmark several frontier LLM agents on simulated biological tasks.

While the ambition to create a scalable, safe benchmark for scientific reasoning is commendable, I am strongly leaning towards rejection. The paper's core methodology is built on a foundation that is fundamentally flawed, making it impossible to draw meaningful conclusions about an agent's true scientific planning capabilities. My primary concerns are: (1) The use of an LLM as a "state transition model" for complex, continuous biophysical processes is profoundly problematic and ungrounded from physical reality. (2) The use of a second LLM as the "evidence classifier" or "judge" creates a closed-loop, self-referential evaluation system that measures alignment with the benchmark's own biases rather than objective scientific success.

**Strengths:**

- clear motivation
- thorough experiments

**Weaknesses:**

- use of LLM as experiment simulator and judge

**Questions:**

- Given that the LLM simulator is ungrounded from physical reality and the LLM judge is self-referential, how can this benchmark measure true scientific capability rather than just alignment with its own internal textual biases?

---

### Official Review · Reviewer_4V36 · 2025-10-31

**Soundness:** 1
**Presentation:** 2
**Contribution:** 1
**Rating:** 2
**Confidence:** 4

**Summary:**

This paper introduces WetBench, an LLM-based simulation environment for evaluating AI systems' ability to design wet-lab experiments. The key idea is to use LLMs as both state transition models (simulating experimental outcomes) and evidence classifiers (judging whether experiments provide sufficient information). The authors create 18 expert-curated experimental tasks across biology domains and benchmarked several frontier models.

**Strengths:**

1. well-motivated. The paper aims to address an important bottleneck in creating AI scientist benchmark, the high cost of generating real-time expeirmental feedback.

2. The paper is well-organized and well-written.

**Weaknesses:**

Major issue:
1. What is this benchmark measuring? The paper claims to evaluate "wet-lab experiment planning and design" but actually measures something quite different. LLMs generate the experimental plans, LLMs simulate the experiment outcomes, and LLMs judge whether the experiments succeeded. This creates a closed loop where we're just measuring how well LLMs agree with themselves or other LLMs, not whether the plans would actually work in a real lab. Even though the result shows that LLM-generated outcomes seem plausible to human experts, this doesn't tell us what the experiments turn out in the real world. There's no comparison with real lab results, no execution of the generated plans, no validation against historical experimental data. Without this grounding, we have no idea if success in this benchmark will translate to success in the real-world.

2. Similarly, the evidence classifier is measuring whether experiments sound convincing to an LLM, not whether they would actuallyb solve the scientific question in a real lab.

3. The paper does not establish the human baseline. How would human experimental biologists score on these same 18 tasks? Without this baseline, we can't interpret whether 44% pass@1 or 72% pass@5 is good or bad.

**Questions:**

see the weakness section

---

### Official Review · Reviewer_gzji · 2025-11-01

**Soundness:** 2
**Presentation:** 1
**Contribution:** 2
**Rating:** 2
**Confidence:** 5

**Summary:**

&nbsp;

The authors introduce WetBench, an LLM-based simulation environment for scientific experiment planning in chemistry and biology. Core features of WetBench include the use of LLMs to generate transitions between individual steps (states) of an experiment as well as the use of LLMs for classifying whether sufficient evidence has been accumulated for the experimental task to be deemed complete. The authors validate the use of LLMs as components of the simulation framework by employing human experts to assess the plausability of the state transitions and evidence classification. I have a number of concerns with the paper in its current form. The largest problem I see with the current work is the overall premise. While it is potentially impactful to investigate the potential of LLMs for experiment planning, I believe a more realistic simulation environment e.g. one grounded in first principles computational techniques would be more useful for benchmarking experiment planning capabilities relative to a framework grounded in LLM-based simulations. Second, I am not convinced by the empirical evaluation. The principal validation of LLMs as simulation components hinge on agreement with human experts, yet no specific details of said experts are presented beyond the fact they comprised "multiple" PhD and postdoctoral researchers. Third, the reproducibility of the current work is highly questionable given that the code is not provided. Lastly, following on from the previous point, the clarity of the work also raises reproducibility concerns. Specifically, the absence of a concrete, mathematical problem definition makes the goal of the current work difficult to follow for the reader, particularly in the early stages of reading the paper. As such, although the topic of the paper is interesting, my recommendation is that the authors revise their simulation framework to feature first principles simulations with LLMs as operators, focus on improving the quality of the empirical evaluation and the clarity of the paper, and submit to a future venue.

&nbsp;

**Strengths:**

&nbsp;

The main strength of the paper is that it considers a problem domain where LLMs can be very impactful, namely in the planning of laboratory experiments in chemistry and biology.

&nbsp;

**Weaknesses:**

&nbsp;

I demarcate between major and minor points below.

&nbsp;

**__MAJOR POINTS__**

&nbsp;

1. **Reproducibility**: The authors could have provided an anonymous GitHub link during the review process or attached the code as supplementary material. Without it, it is very difficult to verify the validity of the results. Additionally, the authors should not claim the open-sourcing of their work as a contribution, since this is false, the project is not currently available.

2. **Data Curation**: For a simulation framework such as the one presented, the authors should document clearly how the dataset was curated and how it will be maintained. An appropriate framework for such data management is to include a datasheet for datasets [2] with the paper.

3. **Clarity**: There is no mathematical problem definition provided in the paepr. Section 3 describes an MDP framework in natural language. It would be beneficial for the authors to formally define this. Such a colloquial presentation of the problem definition is not appropriate for a top tier machine learning conference.

4. **Details of Experiment Design**: There are no details on the expert ratings provided save for the fact that "multiple" PhD or postdoc researchers were selected. How many? What was the split between PhD and postdoc researchers? What were the research specializations of each researcher? What were the exmaples provided to the researchers?

&nbsp;

**__MINOR POINTS__**

&nbsp;

1. There are missing journal/conference/arXiv links for every reference in the paper.

2. There are some missing capitalizations in the references e.g. "LAB-bench" in place of "LAB-Bench".

3. There is no year provided in the references e.g. the first reference should read "Gottweis et al. 2025.

4. Line 68, typo, "LAB-Bench" in place of "Lab-Bench".

5. Line 91, typo, "and identify" in place of "and identifies".

6. In the introduction, for clarity it would help if the authors introduced a mathematical statement of their problem definition. Natural language is ambiguous and as such, the reader will find it difficult to pin down the problem formulation. As an example, in the sentence, "agents submit their evidence to a classifier that determines whether the cumulative results sufficiently justify the experimental goal." what is the scoring mechanism? Is the agent's reward in ${0, 1}$ or in $[0, 1]$? This information is only provided in Section C.3 of the appendix.

7. In the section on related work, it would be worth the authors discussing Bran et al. 2024 [1].

8. Line 151, typo, "materials changes".

9. Line 169, typo, "has access to to conduct its experiments".

10. Line 184, "performed" or "undertaken" would be less colloquial than "done".

11. The format of the appendix is structured incorrectly i.e. the first section on case studies is labelled "B". The section heading A later repeats for "Model Configurations".

12. Line 926, "subtly" is spelled incorrectly in the prompt.

13. The prompts in Section C.2 of the appendix repeat the same message multiple times over 9 pages. For clarity it would have been more beneficial for the authors to highlight what is different across these cases instead of repeating the prompt text.

14. Line 250, typo, "history of observations".

15. Line 1199, typo, "Given actions, materials, and observations".

16. In Figure 3, the meaning of $n$ should be provided in the caption. Presumably it is the number of sampled transitions? If so, why does $n$ differ between the Claude and GPT models?

17. In Table 1, the authors report AVG KAPPA as a metric. What is this metric averaged over? Could the authors please state this in the figure caption.

18. It would be helpful if the authors provided specific examples of type 1 and type 2 errors in Section 5.1.

&nbsp;

**__REFERENCES__**

&nbsp;

[1] M. Bran, A., Cox, S., Schilter, O., Baldassari, C., White, A.D. and Schwaller, P., 2024. [Augmenting large language models with chemistry tools](https://www.nature.com/articles/s42256-024-00832-8). Nature Machine Intelligence, 6(5), pp.525-535.

[2] Gebru, T., Morgenstern, J., Vecchione, B., Vaughan, J.W., Wallach, H., Iii, H.D. and Crawford, K., 2021. [Datasheets for datasets](https://dl.acm.org/doi/fullHtml/10.1145/3458723). Communications of the ACM, 64(12), pp.86-92.

&nbsp;

**Questions:**

&nbsp;

1. In terms of the human evaluators, the confidence score presumably measures how confident the human evaluator is that the state transition is plausible?

2. What do the authors mean when they state, "anecdotal reports suggested that both
models could produce reasonably plausible outcomes"?

&nbsp;

**Details Of Ethics Concerns:**

&nbsp;

No ethical concerns identified.

&nbsp;

---

### Meta-Review · Area_Chair_LTJT · 2026-01-06

**Summary:**

The reviewers agree that while the paper is well motivated and addresses an important bottleneck in evaluating AI-driven scientific experimentation, the proposed benchmark suffers from a fundamental conceptual flaw. By using LLMs to generate experimental plans, simulate experimental outcomes, and judge success, the evaluation becomes a closed-loop, self-referential system that measures alignment with LLM priors rather than genuine wet-lab experimental planning ability. The benchmark lacks grounding in physical or biological reality, provides no human or real-world baseline, and relies on limited, weakly described expert plausibility judgments. These issues are compounded by the absence of released code, unclear dataset curation, and an informal problem formulation, making the results difficult to interpret and the conclusions insufficiently supported for acceptance. Unfortunately, the authors have not provided any rebuttal to further address these concerns. Thus, I think this paper should be rejected.

**Reviewer Concerns:**

The authors have not provided the rebuttal, so none of the concerns are addressed.

**Reviewer Scores:**

All reviewers will keep the original score.

---

### Decision · Program_Chairs · 2026-01-26

Reject